# Participant Perceptions of the Acceptability, Feasibility, and Perceived Impact of the Thiwáhe Gluwáš’akapi Substance Use Prevention Program for American Indian Youth

**DOI:** 10.3390/ijerph22030412

**Published:** 2025-03-11

**Authors:** Raeann L. Vossberg, Monica D. Fitzgerald, Nancy L. Asdigian, Carly Shangreau, Tracy Zacher, Nancy Rumbaugh Whitesell

**Affiliations:** 1Centers for American Indian and Alaska Native Health, University of Colorado, Aurora, CO 80045, USA; monica.d.fitzgerald@cuanschutz.edu (M.D.F.); nancy.asdigian@cuanschutz.edu (N.L.A.); cshangreau@gmail.com (C.S.); nancy.whitesell@cuanschutz.edu (N.R.W.); 2Missouri Breaks Industries Research, Inc., Eagle Butte, SD 57625, USA; tracy.zacher@mbiri.com

**Keywords:** indigenous health, mental health, community, qualitative, cultural adaptation

## Abstract

(1) Background: This study aimed to examine adult participants perceptions of the Thiwáhe Gluwáš’akapi (TG) program. We recruited 13 of 85 (15.3%) adult participants from various previous cohorts of the TG program, separated into lower and higher participation groups. Qualitative semi-structured interviews were conducted. This study was conducted on a Northern Plains reservation, and interviews took place via phone. (2) Methods: Semi-structured interviews with 13 adult participants with children aged 10–12 were completed. Audio files were transcribed and analyzed with ATLAS.ti. (3) Results: Qualitative analysis of these interviews revealed several themes: positive reception by families, enhanced connections to Lakota culture and community, and improved familial relationships. Several barriers to participation emerged, such as difficulties with transportation, scheduling conflicts, and lack of time, which can inform implementation strategies. Observed themes showcase positive impacts of TG on parent–child relationships and cultural connection, aiding overall wellbeing and substance use discourse. (4) Conclusions: Participants in the TG program expressed high satisfaction with the program, gained new skills, and improved family dynamics. Future implementation of TG should include additional transportation support and session scheduling options, in addition to updated implementation strategies to further improve Lakota families’ mental health and wellbeing.

## 1. Introduction

American Indian and Alaska Native (AIAN) populations have been forced to face disruptions in their personal, social, and communal lives in many ways as a result of federal colonization policies, including forced relocation, mandated boarding schools, the adoption of foreign culture at the expense of their own, and prohibitions against Native language use and spiritual practices [1]. The disproportionate burden of illness and substance use in AIAN populations is attributable to these disruptions [1]. The National Survey of Drug Use and Health reported that, as of 2018, 10% of AIAN individuals had a substance use disorder, and 7.1% had an alcohol use disorder [2] and that in 2019, one in five experienced a mental illness [2,3]. Additionally, AIAN youth are at an elevated risk of developing substance use problems [4], among other health conditions and poor mental health outcomes [5]. AIAN youth are also at significant risk of suicide; it is the second leading cause of death among AIAN youth aged 10–24 [6]. Substance use and mental health issues are frequent comorbidities, creating a vicious cycle for AIAN individuals and families [7].

Culturally based interventions can assist in the prevention of poor mental health and substance use outcomes for AIAN youth, serving to reconnect and heal the disruptions AIAN populations have experienced [8]. Though there have been a growing number of culturally based interventions for AIAN communities, there remains a lack of family-based interventions grounded in cultural knowledge. The Thiwáhe Gluwáš’akapi (TG, sacred home in which family is made strong) program was adapted from the Iowa Strengthening Families Program for Parents and Youth 10–14 (SFP 10–14) in partnership with American Indian (AI) community members, a Community Advisory Board (CAB), and program advisors from the original program [9]. The TG program utilizes a family-based approach to engage community members in a multi-faceted process to improve mental health, wellbeing, and substance use outcomes. Community members play a vital role in the TG program, and facilitators are recruited from each community to assist with implementation to ensure cultural competency and improve retention and targeted program outcomes.

The TG project grew from over 20 years of collaboration between the Centers for American Indian and Alaska Native Health at the University of Colorado and a Northern Plains reservation community, focused on mental health and wellbeing. TG infuses the SFP 10–14 with community-specific cultural perspectives and historical sensitivities while maintaining the core structure and lessons of the prevention program [9]. This reservation community has withstood long periods of colonization as part of efforts to systematically eradicate their culture. Such oppression has resulted in high levels of trauma and increased risk for problematic substance use, as well as early initiation of substance use [10]. Results from an initial evaluation of proximal program outcomes indicated that TG holds promise for helping AI families address these risks for youth substance use through positive impacts on risk and protective factors associated with early substance use [10]. In that analysis, youth reported significant improvements in parental communication about substance use, substance use resistance skills, stress management, family cohesion, and overall wellbeing [10]. In addition, adults reported improvements in a wide array of parenting behaviors and positive indicators of family dynamics [10]. A systematic evaluation of long-term program effectiveness is currently being conducted as part of a cluster-randomized controlled trial (CRCT) [11]. The purpose of this qualitative study is to assess the feasibility, acceptability, and impact of the TG program from the perspective of adults enrolled with their children in the CRCT. The insights gained from this analysis are intended to improve the future implementation of the TG program and inform the design and delivery of other substance use and mental health interventions in this population.

## 2. Materials and Methods

### 2.1. Setting

The TG program was implemented in a rural Northern Plains reservation community. Alcohol and most other drugs are illegal on the reservation; marijuana was legalized in 2020 [9]. The tribal Research Review Board overseeing research on the reservation reviewed and approved this study, as did the Colorado Multiple Institutional Review Board.

### 2.2. Participants

Twenty-six adults who participated in the TG program with their 10–13-year-old child(ren) as part of the first three (of four) cohorts of the CRCT were invited to take part in semi-structured interviews with a member of the study team. Members of the study team made phone calls to all twenty-six adults to attempt to recruit them for the qualitative study. These participants had attended in groups based on their location on the reservation, though they were recruited independently for this qualitative study. All enrolled participants were invited regardless of their level of program attendance. However, we divided participants into the following two groups: (1) higher attendance defined as attending at least three of the seven TG sessions, and (2) lower attendance defined as attending two or fewer sessions. Separate interview guides were created for these two groups, with additional questions for low-attendance participants focused on barriers to participation.

### 2.3. Program Implementation

TG was delivered through seven weekly sessions with groups of eight to ten families. Each meeting was led by three to four trained community facilitators and included a shared meal, separate hour-long sessions for parents and youth, and an hour-long family session. Topics included listening and communication, family values, rule setting, consequences and rewards, supporting family members, coping with stress, and resisting peer pressure. Native cultural components were included to complement and reinforce core content.

### 2.4. Recruitment and Interview Process

Adults in each group were contacted by phone in October and November 2023 and invited to take part in a brief phone interview about their perceptions of the program’s acceptability (e.g., the extent to which they enjoyed the program), feasibility (e.g., barriers to participation), and impact on intended outcomes (e.g., family connections and parenting skills).

Interested participants who had time available when initially contacted were immediately consented to (including consent to audio recording) and interviewed. Participants who were interested but unavailable were then re-contacted for the consent process and interview at a more convenient time. All were informed that the interviews would last approximately 30 min and were given a USD 50 gift card for their time.

### 2.5. Analysis

Interview recordings were transcribed via Ubiqus On Demand. The transcriptions were coded for analysis between February and April 2024 by three authors using Atlas.ti following a matrix analysis [10]. Coding of the transcripts utilized thematic analysis, using both a priori and emergent codes in a mixed deductive and inductive approach [10]. The analysis included three authors (R.V., M.F., and C.S.), one from the Tribe and two from the University of Colorado, in order to ensure cultural understanding and sensitivity was achieved alongside ensuring academic rigor. These authors developed the deductive codes prior to the coding process and then coded 15% of the transcripts together to establish consensus and resolve discrepancies. Two authors (R.V. and M.F.) then coded another 15% of the transcripts together to achieve consensus and adjust any final inconsistencies in the codebook to support intercoder reliability. One author (R.V.) then coded the final 70% of the transcripts utilizing the built codebook. Additional inductive codes were added as more transcripts were coded. Following the completion of coding, the quotes and codes were thematically analyzed via matrix analysis by two authors (R.V. and M.F.), condensed into larger themes, and key summary statements were created. The matrix analysis, themes, and summary statements were reviewed by all authors for accuracy and consensus.

## 3. Results

Our final sample of 13 adults included eight higher attendance participants and five lower attendance participants. Adult participants in the CRCT were primarily female (93%), and all interview participants were female. The average age of interview participants was 42.7 years, slightly younger than the average age of all adult participants (48.2 years).

A.Acceptability

Through the qualitative interviews, we hoped to glean information about how acceptable the TG program and its intervention were to participants. One key theme uncovered in our qualitative inquiry was that both higher and lower attendance participants felt the program was very acceptable and enjoyed their experience in the TG program. In the interviews, participants named many aspects of the program that they enjoyed, including games, activities, meals provided, connecting with other members of the community, youth making connections with their peers, developing positive relationships with program facilitators, improving family relationships, and employing more effective communication within their families. In general, participants said they felt the program was a good use of their time and energy, that it was beneficial to them, and met their expectations. Many participants expressed gratitude for the program and overwhelming support for participation in the project. A few adults mentioned that the youth they attended with also enjoyed the program, and many described the games and activities that they or their youth enjoyed.


*“I’m very thankful that whatever brought this to my attention because I got to be part of it and my kids got to be part of it and they learned something from it. I’m really thankful we experienced that.”*


B.Feasibility

Many interviewees discussed key barriers to participation, as well as critiques of the program. A primary critique was wishing more families had participated so they could learn more from each other. A key barrier was the scheduling, which resulted in lower attendance at sessions. Though overall, the adult participants expressed satisfaction with the program’s scheduling and structure, low-attendance adults shared that the timing of the sessions made it challenging for them to attend more frequently. The program has sessions once per week for seven weeks, held on weekday evenings in either the fall or spring. Some participants discussed that having greater flexibility in the session scheduling would make it easier to attend sessions, such as offering sessions in the summer or in the morning instead of the evening, allowing more families the opportunity to attend.

Low-attendance adults in particular provided more insight into the barriers and issues that prevented them from attending more sessions. The most consistently discussed barrier to participation was lack of transportation.


*“[What made it hard to come to the sessions?] Actually, it was just the transportation.”*


Many low-attendance participants stated that they did not have a way to attend the sessions, or if they did, that the distance was often too far for them to travel or that it was too expensive to pay for gas to attend the sessions.


*“Mainly it [difficulties getting to sessions] was the ride because at the time I lived out in the country.”*


Conversely, many participants who were assisted by team members via pickup and brought to sessions stated that provided transportation was one of the program’s biggest strengths. Another barrier often noted was family obligations such as medical appointments, school meetings, sports, community events, and family emergencies or illnesses. Some of these conflicted with session scheduling, and families had to choose between them, preventing them from attending. Additionally, a few participants unfortunately experienced the loss of a family member, friend, or community member, which impacted their ability to attend sessions.


*“Sometimes I would have to go to appointments with my kids or I would have other stuff going on. Even in the evening time, if we had a dinner or, you know, a couple times I missed a couple sessions. My grandma actually passed away during the days of sessions. We had a death in the family so wakes, funerals, that kind of thing made it difficult. The fact that I didn’t have—or my vehicle wasn’t working the best wasn’t even an issue because they provided us with transportation to and from which I was very, very grateful for. And the fact that they had food there, because I was worried about missing dinner. So it worked out.”*


C.Perceived Impact

1.Family Relationships

Another key theme derived from our qualitative study was the impact of the TG program on family relationships. Participants reported learning many new parenting and family skills and gaining important insights into the lives of their family members, which helped improve their relationships. One of the most consistent themes heard in our interviews was the improvement in communication within families in the program. There was a strong emphasis in the program on active listening skills and how to work together as a family. Almost all adult participants mentioned how they feel they can communicate more effectively with their children after the sessions.


*“After the program, we’re talking more. They’re talking more to me… Because before the program, I was doing all the talking. I was talking and sitting here and then we went through this program and now I’m sitting here and they’re the ones coming to me and talking… So after this program, it opened up a barrier probably I guess. I don’t know how to say that. It opened up something. So we’re talking more. I’m more listening more, being more patient.”*


Additionally, parents felt that their youth were also more willing to be open with them in return, leading to a greater understanding of each other and a deeper connection. Many adults in the program also discussed how the program helped them create stronger relationships with their youth through the skills they learned in the program.


*“For me and my son it actually helped us communicate a little better. During the duration of the program, I didn’t really know that there were some things I was doing mean or like how I would talk to my son would sometimes bother him. So one of my biggest things that I learned from the program was my son thought that I was lecturing quite a bit instead of explaining to him. So we had to talk and communicate a little bit better. So it gave me some insight into how to be a better parent, how to go about things in a better way and to talk to him better in a way that he’s able to communicate in a healthy way with me basically.”*



*“I could tell that he [child] enjoyed having me there. So it kind of helped us a bond a little bit better. And I think he enjoyed it as much as I did. I took a lot from it because it helped me be a better parent.”*


Some adults mentioned prioritizing family time and doing things together as a family more often following the program, while others discussed utilizing new skills such as rules and consequences, boundaries, and others. Interviewees also felt their youth were also more willing to listen to them and put effort into their relationships after the program.


*“So one of my favorite things was when they would, after when we’d separate for the classes… then we would get back together, we’d do like little games and stuff over stuff that we’d already learned during that session. It was a really cool way for us to like utilize what we learned, remember it. It helped me put it towards the relationship with my son.”*



*“At first they [youth] were, ‘Why we got to do this?’ but then after a while they got into it and they were actually reminding me [to go to sessions].”*


2.Native Culture and Community

Another aspect of the TG program that is important for healing disruptions and positively impacting mental health and wellbeing is the connection families have to their communities and tribal culture. Many participants discussed the connections they were able to make with other parents in their community because of the program. Many participants felt that the weekly sessions allowed these participants to come together consistently and provide social support in a way that may not typically be available to them. Additionally, several interviewees described how impactful it was to have other parents and caregivers in the program because they were able to learn from one another and help one another with their youth.


*“The most important part were the other adults listening to you. They didn’t like judge you or anything, you know. They tried to help you.”*



*“It [the program] gave us time to interact with each other.”*



*“I felt I was able to share and picking up off the other parents and stuff or just hearing insight from the was eye opening to how, you, know, maybe some of our kids learn in different ways, that they worked with their kid to help me or to give me insight and that helped me deal with my son.”*


The comprehensive view of the program sessions was a warm, comfortable, and supportive environment for adults to connect and form deeper bonds. Additionally, the adults we interviewed reported that their youth also formed deeper connections with friends and peers who attended the sessions. Overall, the participants responded positively to the cultural aspects of the program, and many described particular cultural activities that they felt to be beneficial.


*“I liked [learning] that our culture has more activities than other cultures.”*



*“I liked the presentation of the [Tribal] culture.”*


A few of the participants also described utilizing the tribal language more often in their homes following the program and paying more attention to certain aspects of their culture, such as imagery, drumming, working with their hands, and traditional tribal teachings in their family’s everyday life. A few of the quotes summarizing the impact of the TG program related to community and culture are as follows:


*“I know my language because I know I can still read and write, but like how it goes in every family is like they don’t want to teach their kids the [language], everything that comes with it… But the children just don’t go out and talk, but when they come home they like the words and everything just to make them, I would say, like start out with their sentences and then help them slowly branch out and make a paragraph in the language. So it’s even like the flashcards we did. Those are pretty good. Those are really helpful.”*


D.Substance use

The program also appeared to have an impact on how families conceptualize substance use, and a few adults commented on how substance use has changed for their youths. Parents reported that some youth changed who they hang out with as a result of the program. Several participants also discussed how the program has helped shift their perspectives on substance use, including how they talk with their children about substance use.


*“She’s just not hanging out with kids that are smoking stuff and she’s not doing that anymore.”*



*“She quit hanging around friends that use drugs.”*



*“Maybe [as] someone growing up with alcohol and drugs, knowing that my kids don’t want to do it was helpful to learn.”*


## 4. Discussion

The findings from our qualitative study point to the positive impact the TG program has on families in this community. Adult participants overall felt they gained new skills, became better parents, and had improved relationships within their family and community. They especially highlighted how their relationships with their children improved through better communication, family skills, and spending time and connecting with their youth on a new level. Many participants discussed how they were able to become closer with other adults in the program, and that similarly their youth(s) were able to form deeper relationships with their peers. Many discussed how the program allowed them to connect further with their Native culture by engaging in culturally based activities as well. Additionally, several participants described specific ways that the program has helped them address substance use with their youth and hopefully prevent uptake of substance use. These feelings and experiences showcase the potential of the program to positively impact the mental health and wellbeing of AI families with preadolescent children. Specific outcomes for mental health and substance use are not always directly observable following an intervention or in qualitative research, but the impacts we have gathered thus far indicate positive change is possible through this program.

Lessons learned about feasibility point to the need for the TG program to offer support to communities through additional flexibility in program scheduling. Future implementation should also bolster transportation efforts to ensure more families can make it to sessions and that all families are aware of the transportation assistance available.

### Strengths and Limitations

Though there were many strengths to our qualitative study, there were also several limitations. Overall, we had a small sample size of only 13 participants, which could have limited the perspectives we heard. Though we recruited from three of four cohorts, there was an uneven distribution of the cohorts from which our participants were recruited, as only two were from the spring 2022 cohort, seven from the fall 2022 cohort, and four from the spring 2023 cohort.

Our qualitative inquiry included several strengths, which helped solidify our findings. Our participant sample consisted of a reasonably balanced representation of both lower and higher attendance adults, giving greater context into program participation and barriers. Of the four cohorts of the TG program, we interviewed participants from three of the cohorts, allowing us to confirm similar findings about the program over time. Because of the long-term partnership between the university and the community, we were able to gather additional data from the participants based on the trust fostered prior to this qualitative investigation. Additionally, one of the interviewers and coders is a community member who lives and works in the community, giving us community-based context to the feedback collected.

There were, however, a few limitations as well, including that there were missing data from the interviews, as the transcription software was unable to pick up certain words and sentences from the audio recordings. All our qualitative data were self-reported, which can introduce bias; in particular, recall bias could have occurred as many of the participants were answering questions over a year following their participation in the program. Additionally, our final sample comprised 93% women, and though the larger sample of participants in the TG program is also comprised mainly of women, and our qualitative sample was representative of the larger sample, the perspectives of men are largely missing from this study. Lastly, adults answered several questions on behalf of their youth, and we did not collect that data from the youth directly.

## 5. Conclusions

The next step for the TG program includes both a longitudinal follow-up of CRCT program participants and an implementation study to examine strategies for sustainable implementation in the community. The implementation study will benefit from lessons learned in this study regarding barriers to participation. Some of these include addressing the needs for transportation for participants, increasing the flexibility of program sessions by days of the week and time of day, and exploring alternative formatting (e.g., a weekend retreat format, which was successfully piloted in the summer of 2024) to ensure that more families can participate.

## Data Availability

The datasets presented in this article are not readily available because they belong to the participating sovereign Tribal Nation and any use of these data requires prior approval from the Tribal Research Review Board. Requests to access the datasets should be directed to the Nancy Rumbaugh Whitesell, Principal Investigator of the study.

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
