# Peer review of "Participant Perceptions of the Acceptability, Feasibility, and Perceived Impact of the Thiwáhe Gluwáš’akapi Substance Use Prevention Program for American Indian Youth"

_ijerph, 2025, doi:10.3390/ijerph22030412_

Round 1

Reviewer 1 Report

Comments and Suggestions for Authors

General

·      I applaud the authors for developing and implementing this program. The topic is an important topic and is of interest to Indigenous communities.

·      It would have been helpful to have some additional detail on the program itself such as how adults were selected to participate and if there was a strategy for grouping individuals together.

·      As this is a qualitative study in an Indigenous community, I would suggest that the authors consider including a position statement. 

·      The data that was captured is quite rich, however, I think there may be a better way to present it. I would encourage the authors to return to the data and perhaps discuss it with a vareity of stakeholders, including the community. This may help to find a way to disseminate the results in a meaningful manenr.

Specific 

·      The authors should provide additional detail on the background of the community and place alcohol and substance use in the larger context of colonization and historical trauma. 

·      I liked that the authors separated th participants into two groups (high and low participation), however, I did wonder how they determined the number of sessions needed to be considered high. Was there a precedential study?

·      Given the low number of participants and relatively short length of the interviews, I am unsure why the authors decided to use split coding with consensus checking. It seems quite reasonable for two coders to independently code all of the transcripts. This would provide further certainty of the results and the additional discussion to reconcile may have provided more insight and depth of analysis.  

·      The data would have been better presented using a table to show the codes along with example quotes early in the discussion of the results.

·      Using a table will also reduce the confusion about what exactly the themes are. For example, the header acceptablility appears to be a quote, but then in the description it state that a key theme is that ht participants enjoyed their experience. Then the next header, perceived impact has several sub-headers under it. Thus, it was unclear to me what the themes were.

·      I would recommend that the authors revise the results section so that the quotes are integrated in the theme. Using a storytelling approach aligns with Indigenous cultures and also helps to breathe life into the themes. Alternatively, having several quotes at the begining and then intepretative text would also work. As a reader, I found it difficult to understand the logic/pattern as to when the quotes were inserted. In several instances there was a quote, then a one sentence paragraph interpretation, and then another quote. 

·      There is an extra space on line 215 before “Tribal culture.”

·      The quote in lines 201-11 should be in italics. Also, quotes within quotes are usually signaled by single quotation marks.

·      Depending on the authors that conducting the interview and reviewed the transcripts, there could be additional limitations or strengths. For example, if the researchers have worked closely with the community or are part of the community and the community already trusted them, they may be able to gain additional information from the participants. Alternatively, if the interviewer or reviewer had no prior connection or understanding of the community, they may be disadvantaged in analyzing the data. 

Finally, although I remain a bit unclear about what the exact themes are, I suspect that there may be richer information available. I would encourage the authors to undertake another review to ensure that they have fully captured what the participants shared. For example, the Substance use header incorporates some rich quotes about the importance of peer groups. Somewhat similarly, the Native Culture & Community header is quite broad and may be represented in multiple ways. 

Author Response

  1. It would have been helpful to have some additional detail on the program itself such as how adults were selected to participate and if there was a strategy for grouping individuals together.

We agree that it is important for the reader to understand how participants were recruited to the study. We have added information in the participants section to provide more clarity on participant (see lines 120-124).

  1. As this is a qualitative study in an Indigenous community, I would suggest that the authors consider including a position statement. 

We have included a positionality statement at the beginning of the paper, thank you for this wonderful suggestion! (see lines 33-48)

  1. The data that was captured is quite rich, however, I think there may be a better way to present it. I would encourage the authors to return to the data and perhaps discuss it with a vareity of stakeholders, including the community. This may help to find a way to disseminate the results in a meaningful manenr.

We very much appreciate and share the reviewer’s commitment to including community voices at every stage of the research process.  Although we are unable to accommodate the particular suggestion of this reviewer to expand this input at this point in the project and within the time allotted to respond with a revision, we did include community input throughout all phases of the project, including the interpretation, summary and reporting of the findings from this study.  We clarify the collaboration of the academic and community partners involved in this work in the positionality statement we added to the manuscript (see lines 33-48).  

  1. The authors should provide additional detail on the background of the community and place alcohol and substance use in the larger context of colonization and historical trauma. 

We agree that more information would have allowed for greater understanding and clarity and have added important background information to the introduction section (see lines 80-83).

  1. I liked that the authors separated the participants into two groups (high and low participation), however, I did wonder how they determined the number of sessions needed to be considered high. Was there a precedential study?

Thank you for bringing this point to our attention, we agree that this can be confusing to the reader. To help differentiate we have changed the groups to lower and higher participation (see lines 14, 154-155, 205-206, 213, & 425).

  1. Given the low number of participants and relatively short length of the interviews, I am unsure why the authors decided to use split coding with consensus checking. It seems quite reasonable for two coders to independently code all of the transcripts. This would provide further certainty of the results and the additional discussion to reconcile may have provided more insight and depth of analysis.

Thank you for this insight, we unfortunately cannot reanalyze the data in the time allotted for this revision. However, when we coded originally, we had very high inter-coder reliability (ICR) throughout our coding process. We coded with two members of the academic team as well as one member who lives in the community and is from the Tribe. Through this partnership, we were able to achieve very high ICR and were able to achieve complete consensus after only a small sample of transcripts were coded together. We wanted to ensure that our understanding of the interview content aligned with the views of our community. We added detail on this process (see lines 160-167).

  1. The data would have been better presented using a table to show the codes along with example quotes early in the discussion of the results.

We thank the reviewer for this suggestion. We had originally included such a table in the paper.  However, after reviewing and revising, we realized that it was very, very large and seemed redundant given the quotes woven into the larger paper. To better present the data we have made edits to the formatting of each section and quote. We hope that these edits will make these important comments more clear (see lines 210-211,236, 251, 272-273, 279-282, 287-288, 313, 338, 343, 365, & 373).

  1. Using a table will also reduce the confusion about what exactly the themes are. For example, the header acceptablility appears to be a quote, but then in the description it state that a key theme is that ht participants enjoyed their experience. Then the next header, perceived impact has several sub-headers under it. Thus, it was unclear to me what the themes were.

Similarly to the comment above, we thank the reviewer for this very important point, and we agree that the data could be displayed more clearly. To help rectify this, we have edited the formatting and hopefully streamlined the results to better align with the theses (see lines 210-211,236, 251, 272-273, 279-282, 287-288, 313, 338, 343, 365, & 373).

  1. I would recommend that the authors revise the results section so that the quotes are integrated in the theme. Using a storytelling approach aligns with Indigenous cultures and also helps to breathe life into the themes. Alternatively, having several quotes at the begining and then intepretative text would also work. As a reader, I found it difficult to understand the logic/pattern as to when the quotes were inserted. In several instances there was a quote, then a one sentence paragraph interpretation, and then another quote. 

Thank you very much for this comment, we agree that the way the information was presented could be improved upon. It is important to us to have an understandable flow of information, so we have edited the paper to reflect a better flow of quotes and information (see lines 210-211,236, 251, 272-273, 279-282, 287-288, 313, 338, 343, 365, & 373).

  1. There is an extra space on line 215 before “Tribal culture.”

This has been fixed.

  1. The quote in lines 201-11 should be in italics. Also, quotes within quotes are usually signaled by single quotation marks.

We appreciate pointing this out, but we think that there may have been a mistake in this review, lines 201-211 contain two separate parts, one of which is a quote that is properly notated, and the other is our written accompaniment summarizing the quote. We did notice a lack of additional quotation marks around a quote within a quote nearby and were able to adjust that (line 210) thanks to this comment.

  1. Depending on the authors that conducting the interview and reviewed the transcripts, there could be additional limitations or strengths. For example, if the researchers have worked closely with the community or are part of the community and the community already trusted them, they may be able to gain additional information from the participants. Alternatively, if the interviewer or reviewer had no prior connection or understanding of the community, they may be disadvantaged in analyzing the data. 

Thank you for this point, we were able to add additional strengths to that section expanding upon the long-term relationship with the community (lines 387-390).

  1. Finally, although I remain a bit unclear about what the exact themes are, I suspect that there may be richer information available. I would encourage the authors to undertake another review to ensure that they have fully captured what the participants shared. For example, the Substance use header incorporates some rich quotes about the importance of peer groups. Somewhat similarly, the Native Culture & Community header is quite broad and may be represented in multiple ways. 

Thank you for this review, we agree that it is important to portray all aspects of important information from our study. We are unfortunately unable to complete this given the time allotted for this revision, however, we have edited the paper to hopefully expand on some of the identified themes.

Reviewer 2 Report

Comments and Suggestions for Authors

This article presents findings from a study assessing perceptions of the Thiwahe Gluwasakpi (TG) program. Participants were recruited from previous studies and qualitative data was collected via interviews conducted on a reservation by phone. The study highlighted some important findings related to aspects of the program participants appreciated and barriers to participation. Overall, I think this manuscript has the potential to make a nice contribution to the literature, but I do have some comments below.

The introduction is very brief, i.e., 2 paragraphs. Expanding the introduction to provide more background on the TG program and the needs to improve the TG program would strengthen the manuscript.

The inclusion criteria labels seem slightly misleading. I don’t think that 3 of 7 sessions should be considered high-attendance. I would recommend a higher threshold or a different term.

Information is needed on the characteristics of the community facilitators and the coders. This has become standard in qualitative papers and provides important information about the potential biases and perspectives of those facilitating the sessions and analyzing the data.

What are the implications of the participants being 93% female? Would it be helpful to restrict to only females and change the title and manuscript to focus specifically on female perspectives? If not, then I think more space in the discussion needs to be dedicated to the implications of this highly unbalanced sample.

While the quotes are necessary and helpful, there was some redundancy that could be removed.

Similar to the introduction, the discussion was brief and would benefit from expansion. Participants expressed a lot of important content that is provided in the results but is not synthesized well in the discussion.

Author Response

  1. The introduction is very brief, i.e., 2 paragraphs. Expanding the introduction to provide more background on the TG program and the needs to improve the TG program would strengthen the manuscript.

We agree that additional information would have provided greater clarity to the manuscript and have added important information to the introduction section of the paper (lines 64-67, 70-78, & 80-84).

  1. The inclusion criteria labels seem slightly misleading. I don’t think that 3 of 7 sessions should be considered high-attendance. I would recommend a higher threshold or a different term.

We appreciate this comment and have changed the reference from ‘high’ attendance to ‘higher’ attendance (see lines 14, 154-155, 205-206, 213, & 425).

  1. Information is needed on the characteristics of the community facilitators and the coders. This has become standard in qualitative papers and provides important information about the potential biases and perspectives of those facilitating the sessions and analyzing the data.

We agree with this comment and have added additional information about the community facilitators and the coders. The information about the community facilitators is in the introduction (lines 74-77), and information about the coders is in the analysis section (lines 159-161).

  1. What are the implications of the participants being 93% female? Would it be helpful to restrict to only females and change the title and manuscript to focus specifically on female perspectives? If not, then I think more space in the discussion needs to be dedicated to the implications of this highly unbalanced sample.

Thank you for pointing this out! We would like to clarify that while we recruited any caregiver to the study, we happened to have more women enroll, and it was more likely to be women who attended the sessions with their youth. Because of this, while we attempted to recruit from all original program participants into the qualitative interviews since there were mainly women who participated in the TG program, we also had primarily women who participated in qualitative interviews. We agree that this is an important finding, and we have added this information into the discussion (lines 409-412).

  1. While the quotes are necessary and helpful, there was some redundancy that could be removed.

We thank the reviewer for this input, however after reviewing, we have concluded that each of the quotes we have included serve a specific purpose important to the community and the future of the TG program, so we have elected to keep the quotes in the manuscript for now.

  1. Similar to the introduction, the discussion was brief and would benefit from expansion. Participants expressed a lot of important content that is provided in the results but is not synthesized well in the discussion.

We wholeheartedly agree that more information could have been added to this section, thank you for pointing this out. We have added summarizing information to the discussion section upon review (see lines 365-373).

Round 2

Reviewer 1 Report

Comments and Suggestions for Authors

I appreciate the authors careful attention and response to the comments and suggestions. The further clarification around the methods and positionally were very helpful and now allows readers to better understand the rigor that was undertaken to produce these findings. 

I understand that the time provided to respond to these comments is quite limited, which makes re-analysis difficult. Reviewing the themes may have produced richer data, however, the additions that were made helped to illuminate the findings of the study. 

Reviewer 2 Report

Comments and Suggestions for Authors

The authors have been responsive to all of my comments. This manuscript will make a nice contribution to the field.